# Uncertainty Estimation via Stochastic Batch Normalization

**Andrei Atanov**[1]**, Arsenii Ashukha**[2,3]**, Dmitry Molchanov**[1]**, Kirill Neklyudov**[1]**, Dmitry Vetrov**[1]
[1]National Research University Higher School of Economics, [2]University of Amsterdam,
[3]AI Centre, Samsung Research Russia
`andrewatanov@yandex.ru, ars.ashuha@gmail.com`
`dmolch111@gmail.com, k.necludov@gmail.com, vetrodim@gmail.com`

## Abstract

In this work, we investigate Batch Normalization technique and propose its probabilistic interpretation. We propose a probabilistic model and show that Batch Normalization maximizes the lower bound of its marginalized log-likelihood. Then, according to the new probabilistic model, we design an algorithm which acts consistently during train and test. However, inference becomes computationally inefficient. To reduce memory and computational cost, we propose Stochastic Batch Normalization – an efficient approximation of proper inference procedure. This method provides us with a scalable uncertainty estimation technique. We demonstrate the performance of Stochastic Batch Normalization on popular architectures (including deep convolutional architectures: VGG-like and ResNets) for MNIST and CIFAR-10 datasets.

## 1 Introduction

Deep Neural Networks have achieved state-of-the-art quality on many problems and are successfully integrated in real-life scenarios: semantic segmentation, object detection and scene recognition, to name but a few. Usually the quality of a model is measured in terms of accuracy, however, accurate uncertainty estimation is also crucial for real-life decision-making applications, such as self-driving systems and medical diagnostic. Despite high accuracy rate, DNNs are prone to overconfidence even on out-of-domain data.

The Bayesian framework lends itself well to uncertainty estimation (MacKay, 1992), but exact Bayesian inference is intractable for large models such as DNNs. To address this issue, a number of approximation inference techniques have been proposed recently (Welling & Teh, 2011; Hoffman et al., 2013). It has been shown that Dropout, a well-known regularization technique (Srivastava et al., 2014), can be treated as a special case of stochastic variational inference (Kingma et al. (2015); Molchanov et al. (2017)). Also Gal & Ghahramani (2015) showed that stochasticity induced by Dropout can provide well-calibrated uncertainty estimation for DNNs. Multiplicative Normalizing Flows Louizos & Welling (2017) is another approximation technique that produces great uncertainty estimation. However, such complex method is hard to scale to very deep convolutional architectures. Moreover, recently proposed Residual Network (He et al., 2015) with more than a hundred layers does not have any noise inducing layers such as Dropout. This type of layer leads to a significant accuracy degradation (He et al., 2015). This problem can be addressed by non-Bayesian Deep Ensembles method (Lakshminarayanan et al., 2017), which provides competitive uncertainty estimation, but it requires to store several separate models and perform forward passes through all of them to make prediction.

Batch Normalization (Ioffe & Szegedy, 2015) is an essential part of very deep convolutional architectures. In our work, we treat Batch Normalization as a stochastic layer and propose a way to ensemble batch-normalized networks. The straightforward technique, however, ends up with high memory and computational cost. We, therefore, propose Stochastic Batch Normalization (SBN) — an efficient and scalable approximation technique. We show the performance of our method on out-of-domain uncertainty estimation problem for deep convolutional architectures including VGG-like, ResNet and LeNet-5 on MNIST and CIFAR10 datasets. We also demonstrate that SBN successfully extends Dropout and Deep Ensembles methods.

## 2  METHOD

We consider a supervised learning problem, with a dataset $D = \{(x_i, y_i)\}_{i=1}^N$. The goal is to train the parameters $\theta$ of the predictive likelihood $p_\theta(y \,|\, x)$, modelled by a neural network. To solve this problem stochastic optimization methods with a mini-batch gradient estimator usually are used.

**Batch Normalization**  Batch Normalization attempts to preserve activations of all layers with zero mean and unit variance. In order to do that it uses the mean $\mu(\mathcal{B})$ and variance $\sigma^2(\mathcal{B})$ over the mini-batch $\mathcal{B}$ during training and accumulated statistics on the inference phase:

$$\mathrm{BN}_{\gamma,\beta}^{\mathrm{train}}(x_i) = \frac{x_i - \mu(\mathcal{B})}{\sqrt{\sigma^2(\mathcal{B}) + \epsilon}} \cdot \gamma + \beta \qquad\qquad \mathrm{BN}_{\gamma,\beta}^{\mathrm{test}}(x_i) = \frac{x_i - \hat{\mu}}{\sqrt{\hat{\sigma}^2 + \epsilon}} \cdot \gamma + \beta \qquad (1)$$

where $\gamma, \beta$ are the trainable Batch Normalization parameters (scale and shift) and $\epsilon$ is a small constant, needed for numerical stability. Note that during training mean and variance are computed over a randomly picked batch ($\mu(\mathcal{B})$, $\sigma(\mathcal{B})$), while during testing the exponentially smoothed statistics ($\hat{\mu}$, $\hat{\sigma}^2$) are used. We further address this inconsistency by proposed probabilistic model.

**Batch Normalization: Probabilistic View**  Note from (1) that forward pass through the batch-normalized network depends not only on $x_i$ but on the entire batch $\mathcal{B}$ as well. This dependency can be reinterpreted in terms of mini-batch statistics $\mu(\mathcal{B}), \sigma(\mathcal{B})$:

$$p_\theta(y_i \,|\, x_i, \mathcal{B}_{\backslash i}) = p_\theta(y_i | x_i, \mu(\mathcal{B}), \sigma(\mathcal{B})) \qquad (2)$$

where $\mathcal{B}_{\backslash i}$ is a batch without $x_i$. Due to the stochastic choice of mini-batches during training, for a fixed $x_i$ $\mathcal{B}_{\backslash i}$ is a random variable, so mini-batch statistics can be treated as a random variables. The conditional distribution $p_\theta(\mu, \sigma \,|\, x_i, \mathcal{B}_{\backslash i})$ is the product of two Dirac delta functions, centered at $\mu(\mathcal{B})$ and $\sigma(\mathcal{B})$, since statistics are deterministic functions of the mini-batch, and the distribution of mean and variance given $x_i$ is an expectation over mini-batch distribution. During inference we average the distribution $p_\theta(y|x, \mu, \sigma^2)$ over the normalization statistics:

$$p_\theta(\mu, \sigma | x_i) = \mathbb{E}_{\mathcal{B}_{\backslash i}} \, \delta_{\mu(\mathcal{B})}(\mu) \delta_{\sigma(\mathcal{B})}(\sigma) \qquad p_\theta(y|x) = \mathbb{E}_{p_\theta(\mu,\sigma|x)} p(y|x, \mu, \sigma) \qquad (3)$$

**Connection to Batch Normalization**  In Sec. A we show that during training Batch Normalization (1) performs the unbiased one-sample MC estimation of a gradient of a lower bound to the marginalized log-likelihood (3). Thus, such probabilistic model corresponds to Batch Normalization during training. However, on test phase Batch Normalization uses exponentially smoothed statistics $\mathbb{E}\mu \approx \hat{\mu}, \mathbb{E}\sigma \approx \hat{\sigma}$, which can be seen as a biased approximation of (3):

$$\mathbb{E}_{p_\theta(\mu,\sigma|x_i)} p(y_i|x_i, \mu, \sigma) \approx p_\theta(y|x, \mathbb{E}\mu, \mathbb{E}\sigma)$$

Straightforward MC averaging can be used for better unbiased estimation of (3), however, it is inefficient in practie. Indeed, to draw one sample from the distribution over statistics (3) we need to pass an entire mini-batch through the network. So, to make MC averaging for single test object, we need to perform several forward passes with different mini-batches sampled from the training data. To address this drawback we propose Stochastic Batch Normalization.

**Stochastic Batch Normalization**  To address memory and computational cost of straightforward MC estimation, we propose to approximate the distribution of Batch Normalization statistics $p_\theta(\mu, \sigma \,|\, x_i)$ with a fully-factorized parametric approximation $p_\theta(\mu, \sigma \,|\, x_i) \approx r(\mu)r(\sigma)$. We parameterize $r(\mu)$ and $r(\sigma)$ in the following way:

$$r(\mu) = \mathcal{N}(\mu|\mathrm{m}_\mu, \mathrm{s}_\mu^2) \qquad\qquad r(\sigma) = \mathrm{Log}\mathcal{N}(\sigma|\mathrm{m}_\sigma, \mathrm{s}_\sigma^2) \qquad (4)$$

Such approximation works well in practice. In Sec. B we show that it accurately fits the real marginals. Since approximation no longer depends on the training data, samples for each layer can be computed without passing the entire batch through the network and it is possible to make prediction in an efficient way.

To adjust parameters $\{m_\mu, s_\mu, m_\sigma, s_\sigma\}$ we minimize the KL-divergence between distribution induced by Batch Normalization (3) and our approximation $r(\mu)r(\sigma)$ for each object:

$$D_{\mathrm{KL}}\left( 1/N {\textstyle\sum_{i=1}^N} p_\theta(\mu, \sigma \,|\, x_i) \,\big|\big|\, r(\mu)r(\sigma) \right) \longrightarrow \min_{m_\mu, s_\mu, m_\sigma, s_\sigma}$$

Since $r$ belongs to the exponential family, this minimization problem is equal to moment matching and does not require gradients computation. In our implementation we simply use exponential smoothing to approximate the sufficient statistics of mean and variance distributions. It can be don for any pre-trained batch-normalized network.

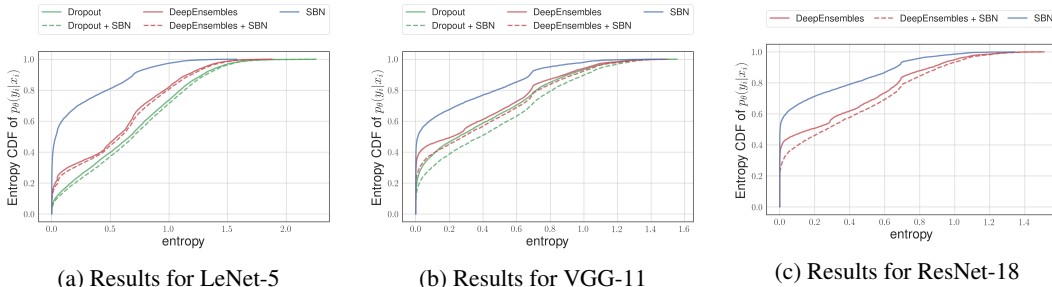

| (a) Results for LeNet-5 | (b) Results for VGG-11 | (c) Results for ResNet-18 |

Figure 1: Empirical CDF of entropy for out-of-domain data. (a) LeNet-5 on notMNIST, (b) VGG-11 and (c) ResNet-18 on five classes of CIFAR10, hidden during training. SBN corresponds to model with all Batch Normalization layers replaced by Stochastic Batch Normalization. The more to the right and the lower, the better.

## 3 EXPERIMENTS

We evaluate uncertainties on MNIST and CIFAR10 datasets using convolutional architectures. In order to apply Stochastic Batch Normalization to existing architectures we only need to update parameters of our approximation $r(\mu), r(\sigma)$ (4), which does not affect the training process at all. We show that SBN improves both Dropout and Deep Ensembles techniques in terms of out-of-domain uncertainty and test Negative Log-Likelihood (NLL), and maintains the same level of accuracy

| Network | Method | Error% | | NLL | |
|---|---|---|---|---|---|
| | | No SBN | SBN | No SBN | SBN |
| LeNet-5 MNIST | SBN | — | $0.53 \pm 0.05$ | — | $0.025 \pm 0.003$ |
| | Deep Ensembles | $0.43 \pm 0.00$ | $0.43 \pm 0.00$ | $0.015 \pm 0.001$ | $0.014 \pm 0.001$ |
| | Dropout | $0.51 \pm 0.00$ | $0.49 \pm 0.00$ | $0.016 \pm 0.000$ | $0.015 \pm 0.000$ |
| VGG-11 CIFAR5 | SBN | — | $5.76 \pm 0.00$ | — | $0.302 \pm 0.002$ |
| | Deep Ensembles | $\mathbf{5.18 \pm 0.00}$ | $5.23 \pm 0.00$ | $0.177 \pm 0.004$ | $\mathbf{0.154 \pm 0.002}$ |
| | Dropout | $\mathbf{5.32 \pm 0.00}$ | $5.38 \pm 0.00$ | $0.155 \pm 0.001$ | $\mathbf{0.149 \pm 0.001}$ |
| ResNet-18 CIFAR5 | SBN | — | $4.35 \pm 0.17$ | — | $0.255 \pm 0.018$ |
| | Deep Ensembles | $\mathbf{3.37 \pm 0.00}$ | $3.34 \pm 0.00$ | $0.138 \pm 0.005$ | $\mathbf{0.110 \pm 0.004}$ |

Table 1: Test errors (%) and NLL scores for known classes. MNIST for LeNet-5 and CIFAR5 for VGG-11 and ResNet-18. SBN column correspond to methods with all Batch Normalization layers replaced by ours SBN.

**Experimental Setup** We compare our method with Dropout and Deep Ensembles. Since He et al. (2015) showed that ResNet does not perform well with any Dropout layer and suffers from instability, we did not include this method into consideration for ResNet architecture. For Deep Ensembles we trained 6 models for all architectures and did not use adversarial training (as suggested by Lakshminarayanan et al. (2017)) since this technique results in lower accuracy.

**Uncertainty estimation on notMNIST** For this experiment we trained LeNet-5 model on MNIST and evaluated the entropy of the predictive distribution on notMNIST, which is out-of-domain data for MNIST, and plot the empirical CDF on Fig. 1(a). We also report the test set accuracy and NLL scores, the results can be seen at Tab. 1.

**Uncertainty estimation on CIFAR10** To show that our method scales to deep convolutional architectures well, we perform experiments on VGG-like and ResNet architectures. We split CIFAR10 dataset into two datasets (CIFAR5) and perform a similar experiment as for MNIST and notMNIST. We trained networks on randomly chosen 5 classes and evaluated predictive uncertainty on the remaining.

We observed that Stochastic Batch Normalization improves both Dropout and Deep Ensembles in terms of out-of-domain uncertainties and NLL score on test data (from the same domain) at the same level of accuracy. However, SBN itself ends up with the more overconfident predictive distribution in comparison to baselines Dropout and Deep Ensembles.

### ACKNOWLEDGMENTS

This research is in part based on the work supported by Samsung Research, Samsung Electronics.

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

## A  LOWER BOUND ON MARGINAL LOG-LIKELIHOOD

In Sec. 2 we propose the probabilistic view on Batch Normalization which models marginal likelihood $p_\theta(y|x)$. In this section we show that conventional Batch Normalization actually optimizes a lower bound on marginal log-likelihood in such probabilistic model. So the goal is to train the model parameters $\theta$ given training dataset $D = \{(x_i, y_i)\}_{i=1}^N$. Using Maximum Likelihood approach we need to maximize the following objective $\mathcal{L}(\theta)$:

$$\mathcal{L}(\theta) = \sum_{i=1}^N \log p_\theta(y_i|x_i) = \sum_{i=1}^N \log \mathbb{E}_{\mu,\sigma \sim p_\theta(\mu,\sigma|x_i)} p_\theta(y_i|x_i, \mu, \sigma) \tag{5}$$

However, the term $\log \mathbb{E}_{\mu,\sigma} p_\theta(y_i|x_i, \mu, \sigma)$ is intractable due to the expectation over statistics. We, therefore, construct a lower bound of $\mathcal{L}(\theta)$ using the Jensen-Shannon inequality:

$$\mathcal{L}_{\text{BN}}(\theta) = \sum_{i=1}^N \mathbb{E}_{\mu,\sigma} \log p_\theta(y_i|x_i, \mu, \sigma) \leq \sum_{i=1}^N \log \mathbb{E}_{\mu,\sigma} p_\theta(y_i|x_i, \mu, \sigma) = \mathcal{L}(\theta) \tag{6}$$

To use gradient-based optimization methods we need to compute gradient of $\mathcal{L}_{\text{BN}}(\theta)$ w.r.t. parameters $\theta$. Unfortunately, distribution over $\mu$, $\sigma$ depends on $\theta$ and, therefore, we cannot propagate gradient through the expectation. However, we can use the definition of $p_\theta(\mu, \sigma|x_i)$ (3) and reparametrize expectation in terms of mini-batch distribution:

$$\begin{aligned}
\mathbb{E}_{\mu,\sigma} \log p_\theta(y_i|x_i, \mu, \sigma) &= \int p_\theta(\mu, \sigma|x_i) \log p_\theta(y_i|x_i, \mu, \sigma) d\mu d\sigma \\
&= \int \left( \int \delta_{\mu(\mathcal{B})}(\mu) \delta_{\sigma(\mathcal{B})}(\sigma) p(\mathcal{B}_{\backslash i}) d\mathcal{B}_{\backslash i} \right) \log p_\theta(y_i|x_i, \mu, \sigma) d\mu d\sigma \\
&= \int \left( \int \delta_{\mu(\mathcal{B})}(\mu) \delta_{\sigma(\mathcal{B})}(\sigma) \log p_\theta(y_i|x_i, \mu, \sigma) d\mu d\sigma \right) p(\mathcal{B}_{\backslash i}) d\mathcal{B}_{\backslash i} \\
&= \int \log p_\theta(y_i|x_i, \mu(\mathcal{B}), \sigma(\mathcal{B})) p(\mathcal{B}_{\backslash i}) d\mathcal{B}_{\backslash i} \\
&= \mathbb{E}_{\mathcal{B}_{\backslash i}} \log p_\theta(y_i|x_i, \mu(\mathcal{B}), \sigma(\mathcal{B}))
\end{aligned}$$

Since distribution over mini-batches does not depend on $\theta$, we now can propagate the gradient through the expectation and use MC approximation for an unbiased estimation. During training Batch Normalization draws mini-batch $\mathcal{B}$ of size $M$ and approximate the full gradient $\nabla \mathcal{L}_{\text{BN}}(\theta)$ in the following way:

$$\nabla \hat{\mathcal{L}}_{\text{BN}}(\theta) = \frac{N}{M} \sum_{i=1}^M \nabla \log p_\theta(y_i|x_i, \mu(\mathcal{B}))$$

Note that Batch Normalization uses the same mini-batch $\mathcal{B}$ to calculate statistics as for gradient estimation. Taking an expectation over mini-batch $\mathcal{B}$, we can actually see that such procedure performs an unbiased estimation of $\nabla \mathcal{L}(\theta)$:

$$\begin{aligned}
\mathbb{E}_{\mathcal{B}} \nabla \hat{\mathcal{L}}_{\text{BN}}(\theta) &= \frac{N}{M} \sum_{i=1}^M \nabla \mathbb{E}_{\mathcal{B}} \log p_\theta(y_i|x_i, \mu(\mathcal{B})) \\
&= N \cdot \nabla \mathbb{E}_{\mathcal{B}} \log p_\theta(y_i|x_i, \mu(\mathcal{B}), \sigma(\mathcal{B})) \\
&= N \cdot \nabla \mathbb{E}_{x_i} \mathbb{E}_{\mathcal{B}_{\backslash i}} \log p_\theta(y_i|x_i, \mu(\mathcal{B}), \sigma(\mathcal{B})) \\
&= \nabla \sum_{i=1}^N \mathbb{E}_{\mathcal{B}_{\backslash i}} \log p_\theta(y_i|x_i, \mu(\mathcal{B}), \sigma(\mathcal{B})) \\
&= \nabla \mathcal{L}_{\text{BN}}(\theta)
\end{aligned}$$

So Batch Normalization produces an unbiased gradient estimation of $\nabla \mathcal{L}(\theta)$ during training and can be seen as an approximation for inference in proposed probabilistic model.

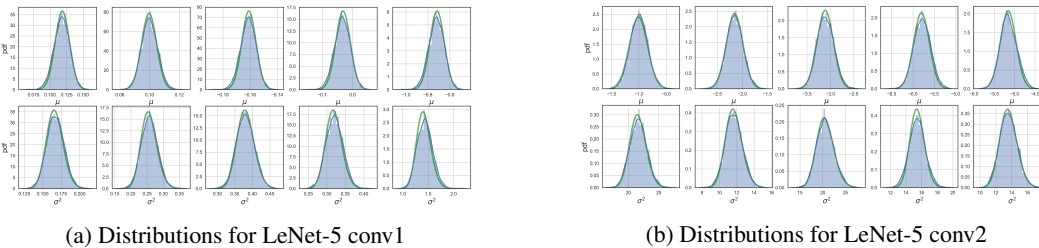

(a) Distributions for LeNet-5 conv1                    (b) Distributions for LeNet-5 conv2

Figure 2: The empirical marginal distribution over statistics (blue) for convolutional LeNet-5 layers and proposed approximation (green). Top row for mean distribution and bottom for variance.

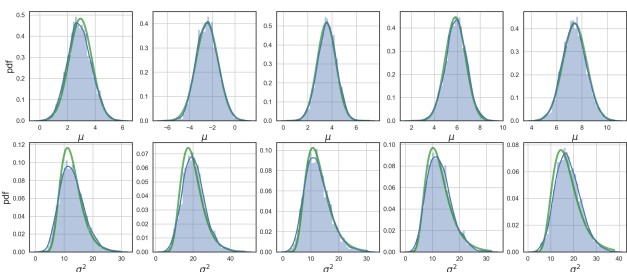

Figure 3: The empirical marginal distribution over statistics (blue) for fully-connected LeNet-5 layer and the proposed approximation (green). Top row corresponds to the means, and the bottom row corresponds to the variances.

## B    STATISTICS DISTRIBUTION APPROXIMATION

For computational and memory efficiency we propose the following approximation for the real distribution over the batch statistics, induced by Batch Normalization:

$$r(\mu) = \mathcal{N}(\mu|\mathrm{m}_\mu, \mathrm{s}_\mu^2) \qquad\qquad r(\sigma) = \mathrm{Log}\mathcal{N}(\sigma|\mathrm{m}_\sigma, \mathrm{s}_\sigma^2) \qquad\qquad (7)$$

According to our observation, the real distributions are unimodal Fig 2. Also the Central Limit Theorem implies that the means converge in distributions to Gaussians, therefore we model this distribution using a fully-factorized Gaussian. While the common choice for the variance is Gamma distribution, we choose the log-normal distribution, as it allows for a more tractable moment-matching. Also as we show in the plots, the log-normal distribution fits the data well.

To verify the right choice of parametric family we estimate an empirical marginal distributions over $\mu$ and $\sigma^2$ for LeNet-5 architecture on MNIST dataset. To sample statistics from the real distribution we pass different mini-batches from training data through the network. We use Kernel Density Estimation to plot the empirical distribution. The results for convolutional and fully-connected layers of LeNet-5 can be seen at Fig. 2 and 3. It can be seen that the approximation (7) fits the real marginal distributions over $\mu$, $\sigma$ very accurately.

