# OpenReview forum: "Uncertainty Estimation via Stochastic Batch Normalization"
_ICLR.cc/2018/Workshop — Accept_

### Official Review · AnonReviewer3 · 2018-03-03
**Interesting interpretation of batch normalization, but more experiments and details needed**

**Rating:** 6
**Confidence:** 3

**Review:**

The authors describe an interpretation of batch normalization (BN) as a probabilistic model with latent stochastic variables. These latent variables are scale and location parameters determined by the randomness in the formation of the minibatch.  After this interpretation, the authors propose to replace the traditional batch normalization method by a probabilistic model which uses the same latent variables but learning a variational approximation for them instead of using the randomness in the minibatch sample. The resulting method is called stochastic Batch normalization (SBN). The authors present different experiments to analyze the performance of SBN. It seems that the main advantage of SBN over BN is that the randomness in the predictions can be computed without having to perform multiple passes of the data through the network.

I think the paper is interesting and relevant (better understanding of batch normalization is necessary by the community), but also needs more work . In particular, I identified the following issues:

1 - The authors do not compare with the original version of batch normalization. It is not clear then what are the advantages of SBN vs. BN.
2 - The results shown do not provide enough evidence of SBN having an advantage with respect to the considered baselines (dropout and ensemble-based methods). More exhaustive experiments are needed.
3 -  From the paper, it is unclear why SBN has a lower computational cost than BN.

---

### Official Review · AnonReviewer2 · 2018-03-09
**Interesting idea but the results are not convincing**

**Rating:** 6
**Confidence:** 3

**Review:**

This work presents a probabilistic perspective on batch normalisation and shows that it maximises the lower bound of a marginalised, over mini batches, log-likelihood. The core idea is that for a given datapoint in the minibatch we can treat the mean and variance statistics induced by the remaining datapoints as a random variable that is integrated out. The authors then proceed to show that during training batch normalization performs an unbiased 1 sample estimate of the bound and then propose stochastic batch normalization (SBN), a technique that allows for efficient Monte Carlo estimation of the average. The overall approach is then evaluated in a variety of experiments that measure performance on in-domain data and uncertainty on out-of-domain data.

The paper presents a simple and interesting idea that can allow for the extraction of uncertainty out of batch normalised networks, which, given the prevalence of batch normalisation in modern networks, can be important. The overall technical presentation is clear although the language in the paper could use more work. Nevertheless, from the experiments it seems that the obtained uncertainty is not very useful as SBN needs to be combined with either dropout or ensembles in order to yield marginal improvements. Furthermore, I think that the experimental sections lacks discussion about the robustness of the results to the minibatch size; to me it seems that this is an important hyper parameter that affects the output uncertainty, since smaller mini batches lead to more noisy statistics and overall higher predictive entropy.

Pros:
	- Interesting idea that allows for the extraction of uncertainty out of batch normalised networks
	- Easy to implement
Cons:
	- SBN alone seems to not lead to particularly useful uncertainty
	- Marginal improvement over dropout and ensembles when SBN is combined with them

---

### Official Review · AnonReviewer1 · 2018-03-11
**An okay interpretation, but needs valid empirical testing.**

**Rating:** 6
**Confidence:** 3

**Review:**

In this paper, the authors provide an interesting probabilistic interpretation to the Batch Normalization trick. Based on the interpretation, the authors provide stochastic batch normalization (SBN) to reduce the memory and computation cost. Specifically, the proposed stochastic batch normalization can be understood as exploiting variational inference technique to approximates the mean and variance. They conducts empirical comparison on both LeNet-5, VGG-11, and ResNet-18 on both MNIST and CIFAR-5 to illustrate the versatile of the SBN.

The only concern I have is about the validation of such Bayesian interpretation of Batch Normalization.

As the authors show in the appendix that the batch normalization is only one variational lower bound of the original MLE, which might not be the optimal bound. Following the variational principle, one can expected better performances if we optimize the distribution of mean and variational to achieve the maximium of the lower bound, rather than just approximating the arbitrary one constructed directlyy from samples. However, the authors directly approximate the distribution via minimizing the KL-divergence between the empirical distribution on mean and variance, rather than optimize the lower bound.

If the probabilistic interpretation is indeed the  reason of the success of the Batch Normalization, following the variational principle should provide better results. Otherwise, such explanation might be not the essential, which may lead to misleading. It will be great to add the empirical study with the approximation from directly minimizing the lower bound to demonstrate the validation of the probabilistic interpretation.

---

### Author Response · Authors · 2018-03-01
**Code for experiments**

Code to reproduce the results of the experiments from this paper is available at https://github.com/AndrewAtanov/stochastic-batch-normalization .

---

### Public Comment · ~Kevin_Smith1 · 2018-05-04
**Request for reference to prior work with significant overlap**

Dear authors, program chairs, and reviewers,

On Oct 27, 2017 our work "Bayesian Uncertainty Estimation for Batch Normalized Deep Networks" was submitted to ICLR and posted on OpenReview.net five months prior to the submission of this work (https://openreview.net/forum?id=BJlrSmbAZ). It also appeared on arXiv on Feb 18, 2018 (https://arxiv.org/abs/1802.06455). The fundamental novelty of interpreting batch normalized networks as probabilistic models is identical in both works. In our article, we made the observation that stochasticity from batch normalization can be exploited to estimate predictive uncertainty. We demonstrated this empirically and argued that this process can be cast as approximate Bayesian inference.

We think the approach proposed in this work is interesting and complementary to our paper but we kindly ask the authors to include a reference to our work along with an appropriate discussion.

Regards,

Kevin Smith (on behalf of Mattias Teye and Hossein Azizpour)

---

> ### Author Response · Authors · 2018-05-07
> **Response**
>
> Dear Prof. Smith,
>
> Thank you for the valuable comment. The works indeed are using similar ideas, but with some significant differences:
>
> 1) We interpret Batch Normalization as a stochastic technique that is not necessarily Bayesian. We also present the real distribution over statistics induced by mini-batches.
> 2) Similar to original Batch Normalization our interpretation is valid without an explicit prior distribution or a regularization term. As a consequence, the training procedure of a batch-normalized network is identical to the optimization of the lower bound on marginal log-likelihood in our model (Appendix A).
>
> We should notice that the papers appeared independently. Nevertheless, your paper indeed appeared at openreview much earlier, and we will cite the paper in our next update.
>
> Regards,
> Andrei Atanov

---

### Decision · Program_Chairs · 2018-03-20
**ICLR 2018 Workshop Acceptance Decision**

**Decision:**

Accept

**Comment:**

Congratulations, your paper was accepted to the ICLR workshop.